# Use of virtual reality in the reduction of pain after the administration of vaccines among children in primary care centers in Central Catalonia: Randomized clinical trial

Mercedes De La Cruz Herrera[1,2], Aïna Fuster-Casanovas[3,4], Queralt Miró Catalina[2,3,4], Mireia Cigarrrán Mensa[5], Isabel Vilanova Guitart[1], Antonio Agüera Sedeño[1], Josep Vidal-Alaball[2,3,4,6]*, Sergi Grau Carrión[7]

**1** Centre d'Atenció Primària Súria, Gerència d'Atenció Primària i a la Comunitat de la Catalunya Central, Institut Català de la Salut, Sant Fruitós de Bages, Spain, **2** Intelligence for Primary Care Research Group, Fundació Institut Universitari per a la Recerca a l'Atenció Primària de Salut Jordi Gol i Gurina, Manresa, Spain, **3** Gerència d'Atenció Primària i a la Comunitat de la Catalunya Central, Institut Català de la Salut, Sant Fruitós de Bages, Spain, **4** Unitat de Suport a la Recerca de la Catalunya Central, Fundació Institut Universitari per a la Recerca a l'Atenció Primària de Salut Jordi Gol i Gurina, Manresa, Spain, **5** VRPharma Immersive Technologies SL, Barcelona, Spain, **6** Faculty of Medicine, University of Vic-Central University of Catalonia, Vic, Spain, **7** Digital Care Research Group, Centre for Health and Social Care Research, University of Vic-Central University of Catalonia, Vic, Spain

* jvidal.cc.ics@gencat.cat

## Abstract

### Background

Pain and anxiety caused by vaccination and other medical procedures in childhood can generate discomfort in both the patient and their parents. Virtual reality is a non-pharmacological distraction technique, capable of diverting patients' attention to a virtual environment, potentially reducing pain signals during the procedure and becoming a method of analgesia, reinforcing traditional methods. The main objective of this study was to evaluate the reduction of pain and anxiety following the administration of two vaccines.

### Patients and methods

A randomized, multicenter, open, parallel and controlled clinical trial was carried out in two assigned groups; in which were included 300 children aged 3 and 6 years, who were candidates to receive two vaccines. The intervention group used virtual reality glasses while the control group received traditional care. Pain and anxiety were assessed using validated scales; the heart rate and the satisfaction of legal guardians were also recorded.

**Data availability statement:** The datasets generated and analyzed during this study contain confidential and sensitive health data from pediatric participants. Publicly sharing these data would breach compliance with patient privacy regulations and the protocol approved by our ethics committee. The IEC/IRB of IDIAP Jordi Gol holds the following accreditations granted by The Office for Human Research Protections (OHRP) - US Department of Health & Human Services Office of the Secretary Office of Public Health & Science Office for Human Research Protections (HHS): Institutional Review Board (IRB): IRB00005101 IRB Organizations (IORGs): IORG0004303 Federalwide Assurance (FWA) for the protection of human subjects for international (Non-US Institutions):FWA Number 00009235 Chairman: Dr. Rosa Morros Email: cei@idiapjgol.org Phone: +34 934 82 41 24

**Funding:** The author(s) received no specific funding for this work.

**Competing interests:** Mireia Cigarrán Mensa, cofounder and CEO of VRPharma Immersive Technologies SL, has an economic interest in the study. To mitigate any potential conflict of interest, VRPharma staff were not involved in data collection, statistical analysis, or the representation of results. This does not alter our adherence to PLOS ONE policies on sharing data and materials.

## Results

A total of 150 patients were included in each group. 53.3% of the intervention group reported no pain during the first vaccination, compared to 25.3% of the control group; with regard to the second vaccination, 49.3% of the virtual reality group reported no pain, compared to 21.3% of the control group. 71.1% of the intervention group faced the first vaccination calmly; 52% and 56.7% did not show anxiety at the end of the vaccination respectively; as for the control group, 71.8% faced the first vaccination calmly, 23.3% did not show anxiety at the end of the first intervention and 19.3% did not show anxiety at the end of the second vaccination. The heart rate was significantly lower in the intervention group. The satisfaction of legal guardians in the virtual reality group was higher than in the control group.

## Conclusions

the use of virtual reality reduces pain and anxiety during vaccination in pediatric population.
Registered in ClinicalTrials.gov: NCT06313762.

## Introduction

Vaccination is a brief procedure usually performed during childhood that generates protection against infectious diseases; it often causes pain, triggering phobias and anxiety in children [1]. Not only can this experience be unpleasant for children, but it can also generate anxiety in parents, who may even hesitate to administer future vaccinations [2].

Possible reactions linked to the stress of vaccination include syncope, hyperventilation, vomiting and convulsions among others. Nineteen percent of paediatric patients report needle phobia [3], which can have a negative impact, such as low adherence to vaccination in children [4]. To try to minimise these reactions, a number of recommendations have been developed during vaccination, e.g., breastfeeding, comfortable positioning of children, being held by a parent or a rapid vaccination technique [5,6].

Distraction is considered a non-pharmacological technique that seeks to reduce pain and anxiety in painful procedures in the paediatric age group [7]; preventing painful stimuli from being transmitted neither to the thalamus, nor to the limbic system, nor to the sensory cortex in an effective manner, focusing attention on external and internal stimuli and not on nociceptive stimuli [6,8]. This distraction can be active/immersive, through manipulation of the environment, or passive/non-immersive, through observation [8]; reducing pain and anxiety to varying degrees [7].

Virtual reality (VR) is a computer-based technique that allows the creation of a simulated environment by means of a sensor device [9,10]. Through the so-called psychological theory of "presence" [9,10]; an interactive environment is created in three dimensions: auditory, visual and tactile [7,9], directing the user's attention to

a pleasant environment [7], through a virtual stimulus, activating cognitive and emotional regions in the brain [7,10]; thus offering a solution in the reduction of pain and anxiety during vaccination [11].

Over the last decade, several studies have been conducted on VR to reduce pain and anxiety in the paediatric population during procedures ranging from dental care, oncology, venous access and prolonged hospital admissions [12,13], all of which have shown great benefits.

The primary objective of this study was to test the efficacy of VR in reducing pain. The secondary objectives were to evaluate the control of anxiety during vaccination through VR, assessing parental satisfaction and monitoring adverse effects during the procedure.

## Methods

### Study design, setting, and sampling technique

A randomized, multicenter, open, parallel and controlled clinical trial was carried out in two assigned groups (the control group and the intervention group) with the participation of five primary care centers of the Catalan Institute of Health, in the territorial management of Central Catalonia – Súria, Callús, Valls de Torrouella, Cardona and Sant Fruitós de Bages-.

The study population was centered on children aged 3 and 6 years, included in the patient registry and attended in one of the participating centers and who, according to the vaccination schedule, were candidates to receive the vaccine corresponding to their age.

To detect a difference of one point between the two groups on the pain level scale, a sample of 150 boys and girls in each group was necessary. To calculate the sample, the Grandària Mostral calculator, GRANMO [14], was used; this, assuming a standard deviation of 3 points, setting an alpha risk of 5%, a power of 80% and an estimated loss to follow-up rate of 5%. A standard deviation of this size was considered while thinking of a moderate dispersion in the responses, since we were seeking to evaluate a non-invasive intervention, such as immersive VR, which could generate diverse responses, depending on the level of immersion and the individual susceptibility of the patients.

### Patient recruitment

Patient recruitment was carried out by a team consisting of a pediatrician and a nurse from the participating CAPs. Families were contacted by telephone to schedule the three- and six-year check-ups, as is currently done. Parents or legal guardians were then informed about the study –its objectives, risks and benefits– and were offered the opportunity to participate. In addition to verbal information, an informative document about the study was provided, which was sent by email, and on other occasions it was delivered before starting the medical check-up.

Simple randomization was performed; patients were distributed in a 1:1 ratio; five sequences were generated, one for each center, in which the numbers 1 (control group) and 2 (intervention group) were randomly assigned. The principal investigator prepared the envelopes with the words control and intervention inside, so that the pediatric nurses, after including a patient in the study, would randomly open the envelope and know the group assignment.

Due to the nature of the study, it was not possible to mask the patients or the health professionals. The guardians of the patients who agreed to be included in the study were given an informed consent, which had to be signed by at least one of the parents or legal guardians. With this document, the signer agreed to inform the other parent or guardian.

### Inclusion and exclusion criteria

Children aged 3 and 6 years, candidates to receive one of the following vaccination schedules were included: Measles, Mumps and Rubella vaccine (MMR) and varicella at 3 years, hepatitis A + diphtheria-tetanus-pertussis-polio at 6 years.

We excluded patients with indication to administer a single vaccine, history of physical or mental illness, such as blindness or deafness; known history of episodes of epilepsy or severe kinetosis; under study or diagnosis of an autism

spectrum disorder, patients with any infection, burn or injury to the face, head or neck that might interfere with the placement of the virtual reality device were also excluded. Additionally, we also excluded patients and companions who did not understand and spoke Catalan and Spanish and cases where there was no legal guardian to sign the informed consent.

### Intervention

The patients in the intervention group used Pico Interactive's Pico G2 VR goggles during the administration of the two vaccines, together with an AOYODKG Android tablet connected to the goggles as a controller. The content of the VR device corresponds to a first-person video designed by the VRPharma team, about an adventure game "NeedleCetamol/ Leia's World". The experience features a virtual avatar Leia and a panda bear, who suffers an accident in a cave and is rescued with the help of Leia's magical powers. The video is two and a half minutes long and is synchronized with the time of vaccine(s) delivery. The content is designed to reassure and relax patients aged 3–11 years who are due to receive one or two vaccines. This is achieved by providing positive emotions through Child Life techniques.

The control group used traditional distractors such as being held by the parent or guardian accompanying them to the consultation, stickers at the end of the procedure or a prize prepared by the parents or guardians from home.

### Data collection and sources of information

One member of the team was responsible for data collection in all participating primary care centers; indicating on the record sheet the patient's age, gender, study group, and intervention performed. The patient's condition and heart rate were recorded before and after each of the vaccinations, regardless of the assigned group. Also, the level of pain perception, level of *anxiety*, adverse reactions and level of satisfaction of the parents or legal guardians were evaluated. Additionally, parents or legal guardians of patients in the intervention group, at the end of the vaccination, evaluated their children's experience with immersive VR.

The intervention group used VR glasses during both vaccinations. First, they were helped to put on the device and then the content of "Leia's World" was briefly explained to them. The control group received traditional distractors.

### Instruments

The Wong-Baker Faces Pain Rating Scale, a validated psychometric tool used in RV studies in the pediatric population, was used to assess pain [15]. Anxiety was assessed by means of the Children's Fear Scale (CFS) and the recording of heart rate before and after each vaccination [16]. Regarding parent or legal guardian satisfaction, a direct question from the CSAT satisfaction scale was asked [17]; parents/legal guardians of patients in the intervention group answered a self-sourced satisfaction questionnaire from the University of Lleida, Faculty of Nursing and Physiotherapy [18].

### Statistical analysis

The information obtained in the study was recorded in a web questionnaire generated with the Microsoft 365 Forms tool, analyzed with the R software, version 4.0.3, and stored on the servers of the Catalan Institute of Health of Central Catalonia [19,20].

An intention-to-treat analysis was performed, analyzing patients according to the group to which they were initially assigned.

Categorical variables were described using absolute frequencies and percentages, while continuous variables were described using mean and standard deviation or median and quartiles.

A T-test was used to compare the values related to pain, anxiety and satisfaction between the two groups. The correlation between the perception of pain and anxiety manifested by the patients and the values recorded by the nurse/pediatrician was evaluated using Pearson's correlation. A significance level of 5% and a confidence interval of 95% were set.

## Ethical considerations

The Institute for Research in Primary Health Care Jordi Gol i Gurina (Barcelona, Spain) ethics committee approval the trial study protocol (CEIm code: 21/132 - P). Study protocol published was published in JMIR Research Protocols [21].

## Results

### Description of the sample

During the period from January 2022 to October 2023, 300 patients aged between three and six years old were recruited. They were subsequently randomized and equally distributed with 150 patients in each group (control and intervention). Likewise, the trial was conducted following CONSORT guidelines (see Fig 1).

Flowchart showing the progress of participants through the phases of the study, according to the CONSORT 2010 guidelines. The diagram includes the number of patients assessed for eligibility, excluded (with reasons), randomized, allocated to intervention groups, lost to follow-up, and included in the final analysis.

The groups were comparable in terms of age and gender, showing no statistically significant differences between them (Table 1). The age distribution did not reveal significant differences between the control and intervention groups (p-value: 0.639), neither did the gender distribution (p-value: 0.419) (Table 4). This ensures an accurate assessment of the impact of immersive VR on the aspects studied, minimizing the influence of age and gender disparities in the analyzed population (Table 1).

There were no post-randomization losses or adverse effects during vaccination. In the intervention group, 30 children (20%) removed the VR glasses before the video ended; however, the second vaccine had already been administered and the data had been collected. One of the reasons given by 28 of these children was that they did not want to continue using the glasses any longer, and two of them reported feelings of distress with the final fireworks.

### Results of the evaluation of pain perception using the Wong-Baker pain scale

The results obtained reveal that immersive VR is an effective intervention to reduce pain during pediatric vaccination.

**First vaccine.** The comparison of the medians of the first vaccine shows a statistically significant variation between the groups (p-value: < 0.001). The median pain in the control group is 2 (Q1:0.5 - Q3:3.5), while in the intervention group it is 0 (Q1:0 - Q3:2.0) (Table 2). Also, 53.3% of children in the immersive VR group experienced no pain compared to 25.3% of controls, with a reduction in pain in 28% of children (Table 2). Likewise, mild pain (scores on scale two and four) was reported by 45.4% in the immersive VR group, compared to the 68.6% in the control group. On the other hand, patients who experienced severe pain (scores on scale six, eight and ten) were considerably fewer in the intervention group (1.4%) than in the control group (6.7%) (Table 2). Finally, there were statistically significant differences between the groups (p-value: < 0.001) (Table 2).

**Second vaccine.** As for the second vaccination, the perception of pain was also lower in the intervention group, with a statistically significant difference (p-value: < 0.001) (Table 2).

Therefore, without a significant improvement in pain compared to the first and second vaccinations, the control group maintained the median of the first vaccination (M:2). In that sense, the median of the intervention group was 2, indicating a slight sensation of pain, but lower than that of the control group. Moreover, the quartiles of the control group (Q1:2.0-Q3:4.0) show high levels of dispersion and pain, while the quartiles of the intervention group (Q1:0-Q3:2.0) show low and concentrated values (Table 2).

Finally, 49.3% of children vaccinated with immersive VR reported no pain, compared with 21.3% who reported no pain during vaccination without immersive VR (Table 2), which was consistent with the first vaccination, with a significant reduction in pain perception. In the same way, 50.0% of the immersive VR group reported mild pain, compared with 66.7% of the control group. Also, the perception of severe pain at the end of the second vaccination was significantly lower in

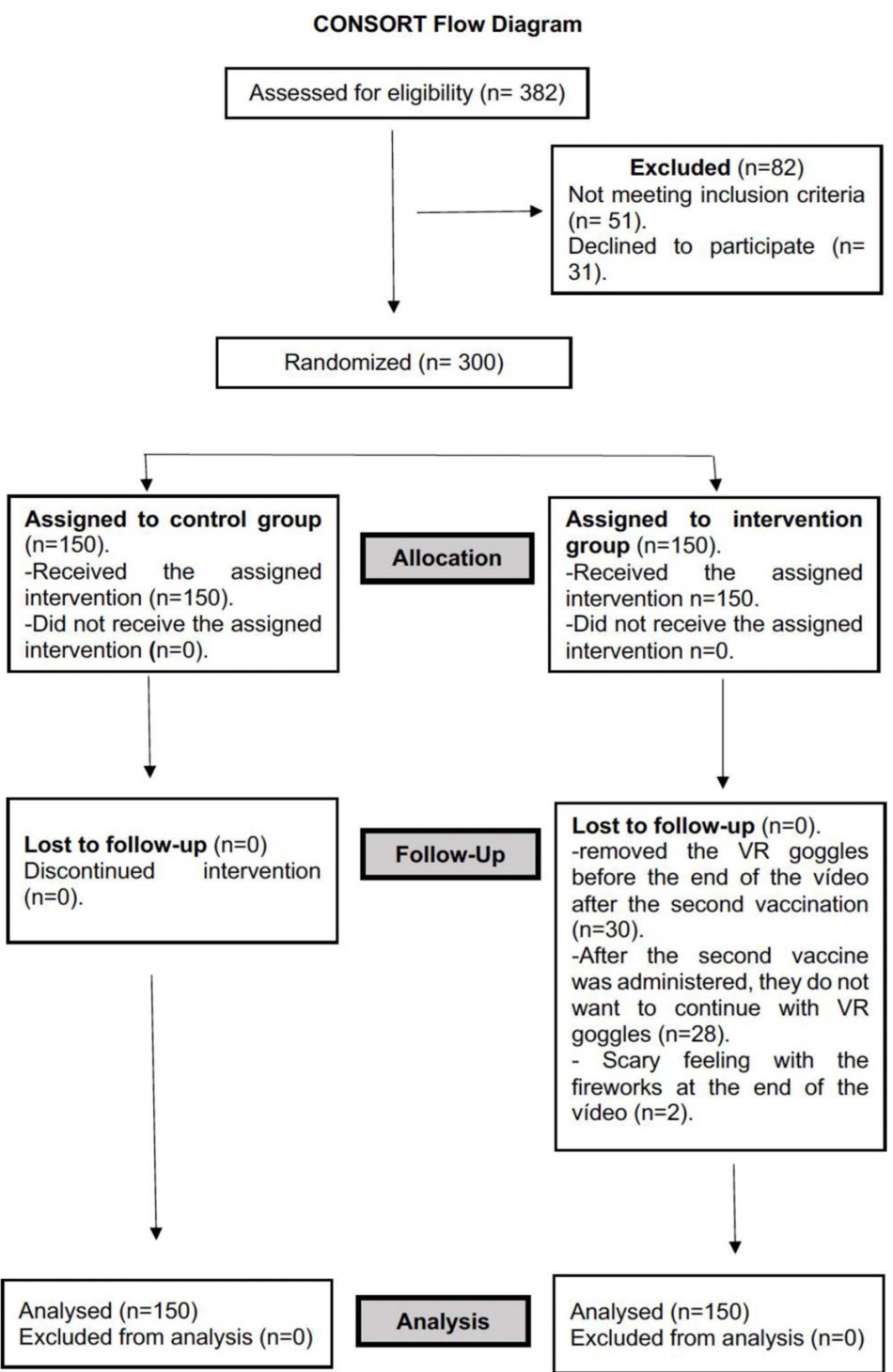

**Fig 1. CONSORT flow diagram.**

**Table 1. Description of the sample of children aged 3 and 6 who have received two vaccines during the review of the Healthy Childhood Program.**

|  | Control (N=150) | VR intervention (N=150) | P-value |
|---|---|---|---|
| **Patient's age (years):** |  |  | 0.639 |
| 3 | 64 (42.7%) | 59 (39.3%) |  |
| 6 | 86 (57.3%) | 91 (60.7%) |  |
| **Patient's gender:** |  |  | 0.419 |
| Child | 78 (52.0%) | 70 (46.7%) |  |
| Girl | 72 (48.0%) | 80 (53.3%) |  |

*VR: virtual reality. Test used: X2*

**Table 2. Description of the results obtained from the measurement of pain using the Wong-Baker facial scale, during the administration of the two vaccines.**

|  | Control (N=150) | VR intervention (N=150) | P-value |
|---|---|---|---|
| **Have you experienced pain during your first vaccination?** |  |  | **<0.001** |
| 0 (no pain) | 38 (25.3%) | 80 (53.3%) |  |
| 2 (mild pain) | 74 (49.3%) | 52 (34.7%) |  |
| 4 (a little more pain than the previous one) | 29 (19.3%) | 16 (10.7%) |  |
| 6 (more pain) | 7 (4.67%) | 1 (0.67%) |  |
| 8 (a lot of pain) | 1 (0.67%) | 1 (0.67%) |  |
| 10 (the worst pain) | 1 (0.67%) | 0 (0.00%) |  |
| **Median [quartiles]** | 2.0[0.5ª-3.5ᵇ] | 0[0ª-2.0ᵇ] | **<0.001c** |
| **Have you experienced pain during the second vaccination?** |  |  | **<0.001** |
| 0 (no pain) | 32 (21.3%) | 74 (49.3%) |  |
| 2 (mild pain) | 51 (34.0%) | 43 (28.7%) |  |
| 4 (a little more pain than the previous one) | 42 (28.0%) | 22 (14.7%) |  |
| 6 (more pain) | 18 (12.00%) | 8 (5.33%) |  |
| 8 (a lot of pain) | 6 (4.00%) | 2 (1.33%) |  |
| 10 (the worst pain) | 1 (0.67%) | 1 (0.67%) |  |
| **Median [quartiles]** | 2.0[2.0ª-4.0ᵇ] | 2.0[0ª-2.0ᵇ] | **<0.001c** |

VR: virtual reality. Test used: X2. Median [a1r quartile; b3r quartile]. cMann-Whitney between groups.

children in the immersive VR group (0.7%) than in boys and girls in the control group (12%) (Table 2). According to this, significant differences were observed between the groups (p-value: <0.001) (Table 2).

**Results of the anxiety evaluation through the Children's Fear Scale during the administration of the two vaccines**

Anxiety levels measured with the Children's Fear Scale showed a significant decrease in the immersive VR group compared to the control group.

**First vaccine.** At the first vaccination, an equal percentage of boys and girls in both groups (71.1% in the intervention group and 71.8% in the control group) were calm at the beginning of the procedure (Table 3).

There is no statistically significant difference between the two groups, according to the comparison of the medians of the first vaccine (p-value: 0.076). Both groups had the median anxiety of 1.0 with no significant changes. However, the intervention group (Q1:1.0-Q3:1.0) has a smaller dispersion than that of the control group (Q1:1.0-Q3:2.0) (Table 3).

**Table 3. Description of the results obtained from the measurement of anxiety through the Children's Fear Scale during vaccination.**

| | Control (N = 150) | VR intervention (N = 150) | P-value |
|---|---|---|---|
| **FIRST VACCINE** | | | |
| **How did the patient arrive at the consultation?** | | | *1.000* |
| Uneasy | 42 (28.2%) | 43 (28.9%) | |
| Calm | 107 (71.8%) | 106 (71.1%) | |
| **Anxiety level at the start of the vaccine** | | | *0.088* |
| 0 (without anxiety) | 29 (19.3%) | 37 (24.7%) | |
| 1 (mild anxiety) | 76 (50.7%) | 81 (54.0%) | |
| 2 (moderate anxiety) | 30 (20.0%) | 23 (15.3%) | |
| 3 (severe anxiety) | 14 (9.33%) | 5 (3.33%) | |
| 4 (very severe anxiety) | 1 (0.67%) | 4 (2.67%) | |
| **Anxiety level at the end of the vaccine** | | | *<0.001* |
| 0 (without anxiety) | 35 (23.3%) | 78 (52.0%) | |
| 1 (mild anxiety) | 57 (38.0%) | 44 (29.3%) | |
| 2 (moderate anxiety) | 38 (25.3%) | 13 (8.67%) | |
| 3 (severe anxiety) | 18 (12.0%) | 12 (8.00%) | |
| 4 (very severe anxiety) | 2 (1.33%) | 3 (2.00%) | |
| **Median [quartiles]** | 1.0[1.0ᵃ-2.0ᵇ] | 1.0[1.0ᵃ-1.0ᵇ] | *<0.076c* |
| **SECOND VACCINE** | | | |
| **How did the patient arrive at the consultation?** | | | *<0.001* |
| Uneasy | 73 (48.7%) | 37 (24.7%) | |
| Calm | 77 (51.3%) | 113 (75.3%) | |
| **Anxiety level at the start of the vaccine** | | | *<0.001* |
| 0 (without anxiety) | 29 (19.3%) | 88 (58.7%) | |
| 1 (mild anxiety) | 46 (30.7%) | 29 (19.3%) | |
| 2 (moderate anxiety) | 43 (28.7%) | 15 (10.0%) | |
| 3 (severe anxiety) | 25 (16.7%) | 12 (8.00%) | |
| 4 (very severe anxiety) | 7 (4.67%) | 6 (4.00%) | |
| **Anxiety level at the end of the vaccine** | | | *<0.001* |
| 0 (without anxiety) | 29 (19.3%) | 85 (56.7%) | |
| 1 (mild anxiety) | 45 (30.0%) | 28 (18.7%) | |
| 2 (moderate anxiety) | 38 (25.3%) | 20 (13.3%) | |
| 3 (severe anxiety) | 30 (20.0%) | 8 (5.33%) | |
| 4 (very severe anxiety) | 8 (5.33%) | 9 (6.00%) | |
| **Median [quartiles]** | 1.0[1.0ᵃ-2,0ᵇ] | 0[0ᵃ-1.0ᵇ] | *<0.001c* |

VR: virtual reality. Test used: X2. Median [a1r quartile; b3r quartile]. cMann-Whitney between groups.

Before the first vaccination, 24.7% of children in the intervention group reported no anxiety at all, compared with 19.3% in the control group (Table 3). Mild to moderate anxiety was reported by 15.3% and 54% of the intervention group, respectively, and very severe anxiety was reported by 2.67% at the start of the vaccination, compared with 0.67% of children in the control group (Table 3). In this case, the differences between groups were not statistically significant (Table 3).

Also, 52% of children who used the immersive VR device completed the first vaccination without anxiety (Table 3). Mild to moderate anxiety was reported by 38% and 25.3% of children in the control group, respectively (Table 3).

At the end of the procedure, severe and very severe anxiety were reported by 2% and 8% of the immersive VR group, respectively (Table 3); this, compared to the control group, which presented severe and very severe anxiety in 12% and 1.33% of the cases (Table 3).

In this process, data were obtained with significant statistical differences in the two groups regarding the anxiety state of the children at the end of the administration of the first vaccine (Table 3).

**Second vaccine.** Regarding the second vaccination, 75.3% of children in the intervention group started the procedure calmly, compared to 51.3% in the control group; therefore, there were significant differences between the groups (p-value: <0.001) (Table 3).

The difference between the medians is statistically significant (p-value: <0.001). The median of the control group is 1.0, while the median of the intervention group is 0. The control group showed greater variability and higher levels of anxiety (Q1:1.0-Q3:2.0), compared to the intervention group, which showed more concentrated values and less variability (Q1:0-Q3:1.0) (Table 3).

Mild-moderate anxiety in the immersive VR group was manifested at the start of the second vaccination in 19.3% and 10% of patients, respectively (Table 3). Also, children in the control group who used traditional distractors presented mild-moderate anxiety at the start of the second vaccination in 30.7% and 28.7% of the cases, respectively (Table 3).

On the other hand, between 4% and 8% of the immersive VR group presented severe and very severe anxiety at the start of the second vaccination (Table 3). Regarding the control group, between 4.67% and 16.7% showed severe and very severe anxiety at the start of the second vaccination (Table 3). That is, statistically significant differences were observed between the groups (p-value: <0.001) (Table 3).

Moreover, between 5.33% and 6% of the immersive VR group presented severe-very severe anxiety at the end of the procedure (Table 3). In contrast, between 5.33% and 20% of the control group showed severe-very severe anxiety at the end of the second vaccination (Table 3). In that sense, the differences between the groups were statistically significant (p-value: <0.001) (Table 3).

## Results of heart rate evaluation during the administration of the two vaccines

HR was measured before and after each of the vaccines to assess the physiological response of pain and anxiety.

**First vaccine.** At the start of the first vaccination, HR medians were similar between the intervention (113 bpm), (Q1:100-Q3:123) and the control group (112 bpm), (Q1:98-Q3:128) (Table 4). Both of them had a similar distribution before the intervention, with relatively wide HR ranges, suggesting normal variability in the anticipated stress response (Table 4). However, no statistically significant differences were observed between the two groups (p-value: 0.244) (Table 4).

At the end of the vaccination, a significant difference in HR was observed in the immersive VR group (median: 105 bpm) (Q1:98-Q3:120) compared to the control group (median: 118 bpm) (Q1:104-Q3:138) (Table 4). Following the first

**Table 4. Description of the results obtained from the measurement of heart rate before and after the administration of the two vaccines.**

| | Control (N = 150) | VR intervention (N = 150) | P-value |
|---|---|---|---|
| **Heart rate** | | | |
| Before the first vaccine | 112[98[a]-128[b]] | 113[100 [a] -123[b]] | *0.244c* |
| After the first vaccine | 118[104 [a] -138[b]] | 105[98[a]-120[b]] | *<0.001c* |
| [d]**p-value between groups** | *<0.001* | *<0.001* | |
| Before the second vaccine | 121 [106 [a] -138[b]] | 104 [97[a] -118[b]] | *<0.001c* |
| After the second vaccine | 128 [110 [a] -150[b]] | 101 [92[a] -119[b]] | *<0.001c* |
| [d]**p-value between groups** | *<0.001* | *0.387* | |

VR: virtual reality. Test used: X2. Median [a1r quartile; b3r quartile]. cMann-Whitney between groups.

vaccination, the range from Q1 to Q3 in the immersive VR group was narrower, indicating that most patients had a more uniform and lower response to stress (Table 4). The lower variability in the immersive VR group suggested a more controlled and less intense response compared to the control group (Table 4). Finally, within the groups, the change in HR was statistically significant (p-value: <0.001) (Table 4).

**Second vaccine.** HR, as at the end of the first vaccination, was significantly lower at the beginning and at the end of the second vaccination in the immersive VR group compared to the control group.

At the start of the second vaccination, the HR medians showed statistically significant differences (p-value: <0.001) (Table 4). In that sense, the intervention group recorded a median HR of 104 beats/min (Q1:97-Q3:118); as for the control group, this recorded a median HR of 121 beats/min (Q1:106-Q3:138) (Table 4). The control group had a higher HR before the procedure, with a wide range of variation, which could indicate an accumulation of anxiety (Table 4). In contrast, the immersive VR group maintained a lower and more controlled range, suggesting that the intervention continued to mitigate stress (Table 4).

At the end of the second vaccination, statistically significant differences were observed (p-value: <0.001) (Table 4). The median of the immersive VR group (101 beats/min) (Q1:92-Q3:119) was significantly lower than that of the control group (128 beats/min) (Q1:110-Q3:150) (Table 4). Likewise, the control group presented a greater dispersion in HR, indicating a more variable response to stress (Table 4). The immersive VR group, however, maintained a narrower and lower range, suggesting that immersive VR not only affects pain perception, but also the physiological response to stress (Table 4). Within the groups, the change over time has been statistically significant (p-value: <0.001) (Table 4).

### Results of the parent/legal guardian satisfaction survey

The children in the control group were accompanied to the vaccination by their mother 82%, by their father 15.3%, and by others 2.7% - maternal or paternal grandparents and/or grandmothers- (Table 5). In the intervention group, 80.7% were accompanied by mothers, 15.3% by fathers, and 4% by others - grandparents and/or maternal or paternal grandmothers - (Table 5).

Sixty-six percent of the parents or legal guardians of the children in the control group were very satisfied at the end of the vaccination process and 34% were satisfied (Table 5). Parents or legal guardians in the intervention group reported 80.7% feeling very satisfied, 18% satisfied, and 13.3% normal after vaccination (Table 5); with significant statistical differences in the two groups.

After vaccination, 99.3% of the parents or legal guardians of the children who used the VR goggles during the procedure considered that the VR goggles helped them to have less pain, 94.7% that their child benefited from their use, 98.7% would allow their use again, and 99.3% would recommend them to other parents (Table 5). Likewise, 32.7% of the parents or legal guardians were aware of VR as an analgesia mechanism in pediatrics, compared to 67.3% who were unaware of it (Table 5).

## Discussion

### Pain and anxiety assessment during vaccine administration

This study evaluated the efficacy of immersive VR as a tool to reduce pain and anxiety during vaccine administration in a primary care setting in children aged three to six years old. The results obtained indicated that children who used immersive VR had less pain and anxiety compared to those in the control group. Such findings that are in line with previous research indicate that immersive VR can be an effective method for distraction and pain reduction in the context of pediatric procedures.

The use of immersive VR as a tool to reduce pain and anxiety in children during vaccine administration is an example of how technology can humanize healthcare. By reducing pain and anxiety, the patient experience is

**Table 5. Degree of satisfaction of parents or legal guardians after vaccination.**

| | Control (N = 150) | Intervention RV (N = 150) | P-value |
|---|---|---|---|
| **Companion Link** | | | 0.812 |
| Others | 4 (2.7%) | 6 (4.0%) | |
| Mother | 123 (82.0%) | 121 (80.7%) | |
| Father | 23 (15.3%) | 23 (15.3%) | |
| **Post-vaccination satisfaction** | | | 0.002 |
| Very Satisfied | 99 (66.0%) | 121 (80.7%) | |
| Satisfied | 51 (34.0%) | 27 (18.0%) | |
| Normal | 0 (0.0%) | 2 (13.3%) | |
| **Do you consider that virtual reality helps to have less pain during vaccination?** | | | |
| No | | 1 (0.7%) | |
| Yes | | 149 (99.3%) | |
| **Do you feel that your child has benefited from the use of VR during vaccination?** | | | |
| No | | 8 (5.3%) | |
| Yes | | 142 (94.7%) | |
| **Would you allow your child to use VR again during vaccination?** | | | |
| No | | 2 (1.33%) | |
| Yes | | 148 (98.7%) | |
| **Were you aware of the usefulness of VR as an analgesia/distractor in pediatric procedures?** | | | |
| No | | 101 (67.3%) | |
| Yes | | 49 (32.7%) | |
| **Would you recommend to other parents the use of VR in the vaccination of their children?** | | | |
| No | | 1 (0.7%) | |
| Yes | | 149 (99.3%) | |

VR: virtual reality. Test used: X2

improved, providing a safer and more comfortable environment. This intervention not only helps pediatric patients cope with medical procedures with less pain and anxiety but also alleviates the emotional burden on parents and medical staff.

Nilsson and Piskorz, in their respective studies, suggest that the analgesic effect of distraction through VR could be greater in patients with underlying pathology, who could better compare the degree of pain perceived in previous experiences without VR [22,23]. The patients in the study were healthy children aged three to six years old who attended a routine pediatric check-up, during which they received vaccines according to their age and vaccination schedule. In that sense, the vast majority of children had been previously vaccinated and were able to compare the pain caused by a vaccination with or without the help of immersive VR [21].

Episodes of increased pain during needle-related procedures have been described in the literature [24]. Around 78% of patients experience pain during hospital stays associated with venous punctures or catheterizations [25]. Orenius et al. concluded that about 21–75% of the sample of pediatric patients studied had a fear or phobia of needles [26]; this coincides with the results of Wong et al., who detected a high incidence of pain during vaccination, intravenous catheterization, administration of intramuscular or subcutaneous medication, and lumbar puncture [27].

These studies reflect a challenging reality for pediatric patients: pain and anxiety are significant components of medical experiences, especially in procedures involving needles. Therefore, pain that is not adequately managed can have long-lasting effects, including phobia of medical procedures and avoidance of health care in the future.

Several studies have demonstrated the benefits of VR in reducing pain and anxiety, with results consistent with those obtained in this study. These studies cover different settings and have focused especially on the hospital setting, with particular attention to pediatric oncology. For example, VR has been used during procedures such as venipuncture and lumbar puncture in cancer patients, where a significant reduction in pain and anxiety has been observed [22,28]. In addition, VR has shown efficacy in pediatric emergency situations [29], as well as in the treatment of burns in both adults and children [30,31]. Its use during dental procedures has also been explored, with positive results in reducing pain and anxiety [32].

On the other hand, evidence for the use of immersive VR in immunization is limited. Studies conducted to date are very diverse in terms of patient type and procedures performed [33].

What is notable about the present study is that the sample is focused entirely on primary care, including patients from two rural primary care centers. This sample specificity is significant, as most of the existing literature focuses on hospital or specialized settings. Therefore, the results of this study not only confirm the benefits of immersive VR in reducing pain and anxiety but also expand existing knowledge by including a less studied pediatric population.

Bergomi et al. [34] report that nonpharmacological methods involving different senses alter pain perception, reduce patient stress, and achieve greater adherence to procedures and treatments. Furthermore, they describe that non-pharmacological strategies distract the patient during venipuncture, decreasing the sensation of pain and anxiety, both in patients and their parents [35], thus achieving positive effects, without adverse effects being observed [36] and, in case of being admitted, the hospital stay is reduced [37].

During the administration of the two vaccines, a relaxation and distraction technique has been used through visual stimulation as a non-pharmacological mechanism. This strategy produced a statistically significant reduction in pain and anxiety in pediatric patients; therefore, it confirms the effectiveness of sensory distraction methods for the treatment of pain in pediatric settings, providing a safe and useful alternative to pharmacological treatments. In addition, the adoption of these strategies can improve the entire experience of patients and their families, which translates into a more favorable perspective of medical procedures and greater collaboration and adherence to health programs in the pediatric age group.

Martin et al. [38] assessed pain and anxiety during venipuncture and peripheral line cannulation in the pediatric emergency department [38]. As in this study, they used the Wong-Baker scale to measure pain and the Groningen Distress Scale to measure anxiety. The results showed no statistically significant differences [38]. In a multicenter randomized clinical trial, Gil Piquer et al. tested the efficacy of VR in reducing pain and anxiety during scheduled blood draws in both hospital and primary care settings, using the visual analog scale for pain assessment and the Groningen Distress Scale for anxiety, obtaining a statistically significant reduction in pain [39].

In Singapore, a pilot randomized controlled trial assessed pain and anxiety during vaccination in primary care for children between 4 and 10 years old [40]. The trial compared outcomes between a control group and an immersive VR group using the Wong-Baker scale for pain and the Children's Fear Scale for anxiety. The results were similar to those obtained in the current study, both indicating a marked decrease in pain and anxiety during childhood vaccination using immersive VR. In that sense, the consistency of the results across different studies and geographic locations underlines the robustness and widespread applicability of immersive VR as a pain and anxiety management tool in pediatrics.

The fact that the present study has a significantly larger sample (300 patients) than the aforementioned studies further strengthen the validity of the results. A larger sample not only provides more representative data, but also allows for more robust statistical analysis, increasing confidence in the conclusions. This is particularly important in pediatric studies, where individual variations in pain and anxiety perception can be wide.

Furthermore, the use of standardized tools, such as the Wong-Baker pain scale and the Children's Fear Scale, facilitates comparison between studies and reinforces the evidence for the effectiveness of immersive VR. These scales are

widely recognized and used in clinical research, ensuring that pain and anxiety assessments are objective and consistent [38–40]. Regarding anxiety in pediatrics, there are no validated scales for children under five years old; however, the Children's Fear Scale has been used in studies in three- and four-year-old children, with results similar to those found in the literature, being used at five years old [16]. To reduce bias in the measurement of anxiety, an additional methodology was used in this study: the children's HR was recorded before and after each vaccine. HR is an objective physiological indicator that can accurately indicate children's stress and anxiety levels.

This complementary technique allowed for a more thorough examination of the emotional responses of the children participating in the study, providing quantitative data to support the subjective observations collected using the rating scales. The use of both subjective and objective measures improves the validity of the results and provides a more complete picture of the impact of the interventions delivered.

Dahlquist et al. stated that active distraction, in which the patient is immersed in a specific experience, is more beneficial than passive distraction with films in the non-pharmacological treatment of pain and anxiety [41]. This approach was validated in the present study, in which VR was used to create an immersive world that removed children from the uncomfortable scenario of vaccination. The adapted graphical experience, such as that offered by the "Leia's World" application, created specifically for the administration of one or two vaccines, provides a visual and mental separation that allows children to focus on an independent narrative, reducing the stress related to the procedure.

This technique is consistent with the findings of Chang et al., who developed a VR program called SILVER (Soothing Immunization Leveraging on Virtual Reality Experience) to increase adherence to seasonal influenza vaccination among the pediatric primary care population in Singapore [40]. SILVER showed significant benefits in pain and anxiety control during vaccination, lending credence to the idea that well-developed, situation-specific immersive VR experiences are useful tools for pain and anxiety management in pediatric medical procedures.

Both findings highlight the advantage of VR in immersive active distraction over more typical passive distraction strategies. As a matter of fact, full immersion in a virtual environment can completely captivate the attention of pediatric patients, allowing them to tune out the pain and anxiety associated with medical procedures. This not only improves the current patient experience but may also increase future willingness to participate in medical care as a result of less traumatic and more positive encounters.

The studies by Özapl et al. and Dumoulin et al. present interesting and contrasting findings on the use of immersive VR as a tool for pain and anxiety management in pediatric medical procedures [13,42]. In the case of Özapl et al., they evaluated the effect of two different immersive VR applications, such as VR-Rollercoaster and VR-Ocean Rift, obtaining mild pain values in the group of patients who used immersive VR; this, compared to the control group who reached moderate pain levels, without finding statistically significant differences in the two groups that performed immersive VR [13]. This suggests that the effectiveness of immersive VR may vary depending on the specific application and the context of the medical procedure.

On the other hand, the study by Dumoulin et al. highlights the effectiveness of immersive VR in reducing pain, compared to the use of a conventional distractor such as television, using the Child Life program, with a statistically significant decrease in the fear of pain, without affecting the intensity of the pain [42]. The graphic experience used in our study is based on the Child Life emotion control technique, which ensures that both the patient and their family adapt optimally to the hospital and disease process, in addition to promoting their emotional well-being. The significant decrease in fear of pain in the immersive VR group highlights an important aspect of the use of this technology: its ability to influence the emotional perception of patients, although not necessarily the intensity of the pain experienced.

In that sense, the literature emphasizes the need for comprehensive approaches that not only address physical pain, but also the emotional aspect in medical procedures. This includes creating a hospital environment that considers the psychological well-being of patients, using distraction techniques, psychological support, and emerging technologies such as immersive VR to improve the patient experience.

## Assessing parent/legal guardian satisfaction during the administration of the two vaccines

Parent/legal guardian satisfaction is a crucial indicator of the success of any pediatric intervention. In the case of immersive VR, parent/legal guardian satisfaction was significantly higher in the pediatric patient group that experienced this technology, results that are consistent with those obtained in other studies [38,40].

Regarding this, Chan et al. conducted a study in Australia that sought to verify the impact of using immersive VR as a mechanism to reduce pain during venipuncture and the perception of parents during the procedure, obtaining lower pain intensity and parents satisfied with immersive VR [43].

Evidence suggests that immersive VR is not only effective in reducing pain and anxiety in children during medical procedures but also improves parent or legal guardian satisfaction. This dual benefit is critical to the acceptance and adoption of new technologies in pediatric healthcare settings. Parent satisfaction is essential, as they are the primary decision makers regarding their children's healthcare and their perception can influence future decisions about their health.

A relevant example is a recent study conducted in a hospital setting, focusing on pre-operative anxiety control specifically using the Doc McStuffins: Doctor for a Day Virtual Reality Experience (DocVR) program [44], which used an immersive VR experience designed to transform a potentially frightening situation, such as preparing for surgery, into a more friendly and relaxing experience.

Successful implementation of advanced technologies such as immersive VR in the healthcare setting can increase trust in the care their children receive. This study shows that parents perceive immersive VR not only as an effective tool, but also as an indicator of commitment in the primary care setting to innovation and the well-being of the pediatric population.

Likewise, patient satisfaction is often a success metric in healthcare institutions; in this case, it is the satisfaction of parents or legal guardians, who continue to be key to the implementation of this type of device in daily clinical practice.

## Safety assessment during the administration of the two vaccines

During the immunization process, no adverse reactions were observed in any of the study groups, which demonstrated a satisfactory safety profile for the use of immersive VR. Safety concerns about immersive VR in pediatric procedures frequently center on the physical fit of the devices, particularly the comfort and appropriateness of the VR headsets for young children. In this study, great care was taken to ensure that the VR headsets were properly fitted, eliminating any discomfort or undue pressure.

In addition, precautions were taken to create a safe environment, reducing the likelihood of accidents with real-world objects while patients were engrossed in the immersive VR experience. These precautions are critical to avoid potential mishaps caused by patient confusion or limited mobility when visually separated from the physical environment. In that sense, the combination of proper device fit and a controlled environment ensures that the immersive VR experience is not only helpful to alleviate pain and anxiety, but also safe for pediatric patients.

Safety is a major concern when it comes to using immersive VR and other technologies in pediatric settings. The findings of the current study, along with the existing body of research, suggest the safe use of immersive VR, with few adverse effects and no appreciable dangers to patients' general or ocular health. These results support the feasibility of incorporating immersive VR into routine clinical practice, offering pediatric patients a safer and improved medical experience.

## Limitations

The impossibility of blinding participants, because, upon opening the envelope, both the nurse and the legal guardians knew the group to which the patient belonged. To reduce subjectivity, the same person recorded the data for both intervention groups on the data collection sheet.

Regarding the pressure of care and time management in the pediatric consultation and considering that the time to care for patients is limited, the time required to confine participants, to obtain consents and to record data ranged around

the five minutes. This could lead to an even higher care load; therefore, in order to prioritize the quality of care, it was carried out outside of working hours.

Parents or legal guardians and minors who did not speak Spanish or Catalan could not participate in the study. The use of immersive VR is more advantageous if the content played is in their mother tongue, hence the future need to translate the content of the experience into other languages, such as Arabic and English.

Finally, the age range generated another limitation, given that, according to the vaccination schedule at these ages, they receive two vaccines (comparison), and because the use of immersive VR is not recommended for children under two years old.

## Conclusions

This study demonstrates that immersive VR during pediatric vaccination in primary care is an effective intervention to reduce pain and anxiety in children aged 3–6 years old. The significant reduction in pain perception and anxiety levels, as well as in the physiological response to stress indicate that immersive VR can effectively divert children's attention from the painful procedure. Furthermore, it is a feasible and safe intervention, as no adverse effects were found with its use, a crucial aspect for its acceptance and implementation in healthcare settings. Likewise, there is a high degree of satisfaction among parents or legal guardians with the immersive experience as a mechanism of analgesia in everyday procedures, such as pediatric vaccination. The positive impact on patients and their parents or legal guardians can increase adherence to vaccination programs and generate more trust in the healthcare system, thus promoting a more friendly and efficient approach to pediatric care.

## Supporting information

**S1 File. CONSORT checklist.**
(PDF)

**S2 File. CONSORT flow diagram.**
(TIF)

## Acknowledgments

The authors would like to thank the families who accepted their children's participation in the study. We are also grateful to the management team of the ABS of Súria for allowing us to conduct the pilot test in their primary care centers and for their unconditional support. Special thanks to the pediatric teams of the ABS of Cardona - Marta Castillo and Núria Solanas - and of Sant Fruitós of Bages - Ana Isabel Gimenez-. We appreciate VRPharma for their trust in us and their support of research in rural pediatric primary care settings. Special thanks also to the Research and Innovation Unit of the Catalan Institute of Health in Central Catalonia.

## Author contributions

**Conceptualization:** Mercedes De La Cruz Herrera.

**Data curation:** Antonio Agüera Sedeño.

**Formal analysis:** Mercedes De La Cruz Herrera, Queralt Miró Catalina, Antonio Agüera Sedeño.

**Investigation:** Mercedes De La Cruz Herrera, Isabel Vilanova Guitart, Antonio Agüera Sedeño, Josep Vidal-Alaball.

**Methodology:** Mercedes De La Cruz Herrera, Aïna Fuster-Casanovas, Queralt Miró Catalina, Josep Vidal-Alaball, Sergi Grau Carrión.

**Resources:** Mireia Cigarrrán Mensa.

**Software:** Mercedes De La Cruz Herrera.

**Supervision:** Josep Vidal-Alaball, Sergi Grau Carrión.

**Visualization:** Mercedes De La Cruz Herrera.

**Writing – original draft:** Mercedes De La Cruz Herrera.

**Writing – review & editing:** Aïna Fuster-Casanovas, Queralt Miró Catalina, Josep Vidal-Alaball, Sergi Grau Carrión.

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
