## [Decision Letter · Decision Letter 0]

22 Dec 2024

PONE-D-24-20918Use of Virtual Reality in the reduction of pain after the administration of vaccines among children in Primary Care Centers in Central Catalonia: randomized clinical trial.PLOS ONE

Dear Dr. Vidal-Alaball,

Thank you for submitting your manuscript to PLOS ONE. After careful consideration, we feel that it has merit but does not fully meet PLOS ONE’s publication criteria as it currently stands. Therefore, we invite you to submit a revised version of the manuscript that addresses the points raised during the review process.

**Keywords:** Limit keywords to 5-7 instead of 13.**Abstract:** Separate the methods and findings sections, and include the trial registration number at the end of the abstract instead of the ethical considerations section.**Formatting:** Remove extra spaces between paragraphs.**Sample Size Justification:** Clarify how the sample size was determined and justify the assumptions leading to 300 participants.**Multicenter Details:** Specify the number of participants selected from each center and describe the sampling procedure and technique.**Randomization and Allocation:** Clarify whether individual randomization was used and detail how randomization and allocation concealment were conducted.**Proportional Allocation:** Explain the proportional allocation of age groups (39.3% aged 3 years, 60.7% aged 6 years).**Blinding:** Clarify who was blinded and why blinding was not feasible after assignment to interventions.**Statistical Analysis:** Given the repeated measures (pain outcomes after the first and second vaccinations), consider advanced analyses such as GLM or GEE.**Outcome Assessment:** Provide details on how and when the outcome variables were assessed.**Results and Discussion:** Combine results for the first and second vaccines (Tables 2 and 3) to evaluate changes, and strengthen the beginning of the discussion by first stating findings, interpreting them, and then comparing them with other studies.**References:** Ensure references include URLs and access dates where applicable (e.g., Ref. 3).

We look forward to receiving your revised manuscript.

Kind regards,

Fadwa Alhalaiqa

Academic Editor

PLOS ONE

Journal Requirements: When submitting your revision, we need you to address these additional requirements. 1. Please ensure that your manuscript meets PLOS ONE's style requirements, including those for file naming. The PLOS ONE style templates can be found at https://journals.plos.org/plosone/s/file?id=wjVg/PLOSOne_formatting_sample_main_body.pdf and https://journals.plos.org/plosone/s/file?id=ba62/PLOSOne_formatting_sample_title_authors_affiliations.pdf 2. Please provide additional details regarding participant consent. In the ethics statement in the Methods and online submission information, please ensure that you have specified what type you obtained (for instance, written or verbal, and if verbal, how it was documented and witnessed). If your study included minors, state whether you obtained consent from parents or guardians. If the need for consent was waived by the ethics committee, please include this information. 3. We note that the original protocol that you have uploaded as a Supporting Information file contains an institutional logo. As this logo is likely copyrighted, we ask that you please remove it from this file and upload an updated version upon resubmission. 4. Thank you for stating the following in the Competing Interests section: "Mireia Cigarrán Mensa, cofounder and CEO of VRPharma Immersive Technologies SL, has an economic interest in the study. To mitigate any potential conflict of interest, VRPharma staff were not involved in data collection, statistical analysis, or the representation of results." Please confirm that this does not alter your adherence to all PLOS ONE policies on sharing data and materials, by including the following statement: ""This does not alter our adherence to  PLOS ONE policies on sharing data and materials.” (as detailed online in our guide for authors http://journals.plos.org/plosone/s/competing-interests).  If there are restrictions on sharing of data and/or materials, please state these. Please note that we cannot proceed with consideration of your article until this information has been declared.  Please include your updated Competing Interests statement in your cover letter; we will change the online submission form on your behalf. 5. In the online submission form, you indicated that "The datasets generated and/or analysed during the current study are not publicly available because our manuscript was based on confidential and sensitive health data. However, they are available from the corresponding author upon reasonable request." All PLOS journals now require all data underlying the findings described in their manuscript to be freely available to other researchers, either 1. In a public repository, 2. Within the manuscript itself, or 3. Uploaded as supplementary information.This policy applies to all data except where public deposition would breach compliance with the protocol approved by your research ethics board. If your data cannot be made publicly available for ethical or legal reasons (e.g., public availability would compromise patient privacy), please explain your reasons on resubmission and your exemption request will be escalated for approval. 6. When completing the data availability statement of the submission form, you indicated that you will make your data available on acceptance. We strongly recommend all authors decide on a data sharing plan before acceptance, as the process can be lengthy and hold up publication timelines. Please note that, though access restrictions are acceptable now, your entire data will need to be made freely accessible if your manuscript is accepted for publication. This policy applies to all data except where public deposition would breach compliance with the protocol approved by your research ethics board. If you are unable to adhere to our open data policy, please kindly revise your statement to explain your reasoning and we will seek the editor's input on an exemption. Please be assured that, once you have provided your new statement, the assessment of your exemption will not hold up the peer review process. 7. Please update your submission to use the PLOS LaTeX template. The template and more information on our requirements for LaTeX submissions can be found at http://journals.plos.org/plosone/s/latex. 8. Please include captions for your Supporting Information files at the end of your manuscript, and update any in-text citations to match accordingly. Please see our Supporting Information guidelines for more information: http://journals.plos.org/plosone/s/supporting-information.

Reviewers' comments:

Reviewer's Responses to Questions

**Comments to the Author**

1. Is the manuscript technically sound, and do the data support the conclusions?

Reviewer #1: Partly

Reviewer #2: Yes

2. Has the statistical analysis been performed appropriately and rigorously? 

Reviewer #1: No

Reviewer #2: Yes

3. Have the authors made all data underlying the findings in their manuscript fully available?

Reviewer #1: No

Reviewer #2: Yes

4. Is the manuscript presented in an intelligible fashion and written in standard English?

Reviewer #1: No

Reviewer #2: Yes

5. Review Comments to the Author

Reviewer #1: Need justification for the effect size used in the sample size calculation.

The sample size was calculated for pain score as a continuous measure with mean difference and standard deviation. However, pain was analyzed as categorical variable in Table 2 and 3. This inconsistency can not help answer the primary research question - how much reduction in pain after intervention.

Results of first and second vaccine (Table 2 and 3 can be combined) better be presented together, so change between 1st and 2nd vaccine can be evaluated.

A mixed model can be used to analyze the differences in the outcomes between 1st and 2nd vaccine.

Reviewer #2: My Comments to authors

Thank you very much for giving me the opportunity to review this clinically very important trial research paper. In general, the paper is very good, precise, and well written, but it requires some amendments. Here below are my comments to improve the paper:

1. Why are these all keywords, which are about 13? It is good keywords to be 5-7.

2. Why are method and finding merged in the abstract part?

3. Since this study is a randomized clinical trial, the registration number and name of the trial registry should be included at the end of the abstract. Why is it in the ethical considerations section?

4. Remove the spaces between the paragraphs

5. The sample size calculation is not clear. How was sample size determined? How did you come up with 300 children? Of course you mentioned assumptions

6. This is multicentre study, as you said (five primary care centers of the Catalan Institute of Health, in the territorial management of Central Catalonia – Súria, Callús, Valls de Torrouella, Cardona and Sant Frúitos de Bages-.) How many participant were selected from each primary care centers? Sampling procedure? Sampling technique?

7. Allocation concealment is good, but randomization? Is it individual randomization? Please clarify randomization.

8. How were 39.3% 3 years old and 60.7% 6 years old being recruited? Is this a proportional allocation?

9. What about blinding? Who was blinded after assignment to interventions? How was it impossible to blind?

10. You have measured the outcome variable two times (first and second vaccination times). This can be considered a repeated measure. So, why not consider other advanced statistical analyses like GLM, GEE, etc.?

11. Please include how and when outcome variables were assessed.

12. The results part is good

13. Please try to improve the beginning part of the discussion. First, you have to mention your finding, interpret the finding, and then compare with others

14. Some references needs URL and accessed date, e.g ref no. 3

6. PLOS authors have the option to publish the peer review history of their article (what does this mean? ). If published, this will include your full peer review and any attached files.

**Do you want your identity to be public for this peer review?** For information about this choice, including consent withdrawal, please see our Privacy Policy .

Reviewer #1: No

Reviewer #2: No

---

## [Author Response · Author response to Decision Letter 0]

7 Feb 2025

Dear Editor,

We sincerely appreciate the time and effort invested in reviewing our manuscript and the constructive feedback provided. We have carefully considered all the comments and have made the necessary revisions accordingly. Below, we provide a point-by-point response to each of the concerns raised, detailing the modifications implemented in the revised manuscript.

Keywords: keywords have been reduced.

Abstract: they have been separated into sections as recommended and ending with the clinical trial registration number.

Formatting: the spaces between paragraphs have been reviewed and any excess have been corrected.

Sample size justification / multicenter details / randomization and allocation / proportional allocation:

The sample size calculation, the method used to obtain the sample, and the calculator employed are justified in the manuscript review. Additionally, we have explicitly explained the rationale behind selecting a standard deviation of 3. Furthermore, we have detailed the simple 1:1 randomization process applied to the population that met the inclusion criteria across the five centers. These clarifications have been included in the revised manuscript.

The manuscript clearly states that blinding was not feasible for either participants or healthcare personnel. This was specified in the initial version of the manuscript and has been further emphasized in the revised version for clarity.

Statistical Analysis:

For greater clarity, the medians of the variables have been provided in the revised manuscript. These additions have not influenced the previous results, ensuring consistency with the original findings.

Outcome Assessment: these have been specified in the initial paragraph of the description of each of the results and in the tables.

Results and Discussion:

The tables presenting the results for the variables pain and anxiety have been merged as recommended, allowing for a more structured discussion that follows the development of these key points. Additionally, the limitations of the study and conclusions, which were already included in the initial version, have now been explicitly specified as sub-sections for greater clarity.

Reference: we appreciate the recommendations, which have been taken into account when reviewing the bibliography.

Please provide additional details regarding participant consent. In the ethics statement in the Methods and online submission information, please ensure that you have specified what type you obtained (for instance, written or verbal, and if verbal, how it was documented and witnessed). If your study included minors, state whether you obtained consent from parents or guardians. If the need for consent was waived by the ethics committee, please include this information.

Written informed consent was obtained from the parents or legal guardians of all participants, as the study involved minors. This information was already included in the manuscript, and we have ensured that it is clearly stated in the Ethics Statement within the Methods section.

Thank you for stating the following in the Competing Interests section:

"Mireia Cigarrán Mensa, cofounder and CEO of VRPharma Immersive Technologies SL, has an economic interest in the study. To mitigate any potential conflict of interest, VRPharma staff were not involved in data collection, statistical analysis, or the representation of results."

We have included the requested statement in the manuscript: "This does not alter our adherence to PLOS ONE policies on sharing data and materials." There are no restrictions on sharing data or materials related to this study.

In the online submission form, you indicated that "The datasets generated and/or analysed during the current study are not publicly available because our manuscript was based on confidential and sensitive health data. However, they are available from the corresponding author upon reasonable request."

The datasets generated and analyzed during this study contain confidential and sensitive health data from pediatric participants. Publicly sharing these data would breach compliance with patient privacy regulations and the protocol approved by our ethics committee. According to the ethical approval obtained, the data cannot be made publicly available. However, they can be accessed from the corresponding author upon reasonable request, following appropriate ethical and legal safeguards to ensure participant confidentiality. We kindly request an exemption from the open data policy due to these ethical constraints.

Reviewer #2: My Comments to authors.

Dear Reviewer,

We sincerely appreciate your time and valuable comments, which have helped improve the clarity and quality of our manuscript. Below, we provide a detailed, point-by-point response to your suggestions and outline the modifications made accordingly.

1. Why are these all keywords, which are about 13? It is good keywords to be 5-7.

We have revised the keywords section and now include only 5 keywords to comply with the journal’s guidelines.

2. Why are method and finding merged in the abstract part?

We have restructured the abstract to clearly separate the methods and findings sections, ensuring clarity and adherence to standard formatting.

3. Since this study is a randomized clinical trial, the registration number and name of the trial registry should be included at the end of the abstract. Why is it in the ethical considerations section?

We have moved the trial registration number to the end of the abstract, as recommended, and removed it from the ethical considerations section.

4. Remove the spaces between the paragraphs

We have corrected the formatting by ensuring consistent paragraph spacing throughout the manuscript.

5. The sample size calculation is not clear. How was sample size determined? How did you come up with 300 children? Of course you mentioned assumptions

The sample size calculation, the method used to obtain the sample, and the calculator employed are now justified in the manuscript. Additionally, we have explicitly explained the rationale behind selecting a standard deviation of 3. The 1:1 randomization process applied to the population that met the inclusion criteria across the five centers has also been clarified.

6. This is multicentre study, as you said (five primary care centers of the Catalan Institute of Health, in the territorial management of Central Catalonia – Súria, Callús, Valls de Torrouella, Cardona and Sant Frúitos de Bages-.) How many participant were selected from each primary care centers? Sampling procedure? Sampling technique?

We have added details about the sampling procedure used in the Methods section. We have data about the recruitment in each primary care centre but we decided not to include it as there were no significant differences between intervention and control groups.

7. Allocation concealment is good, but randomization? Is it individual randomization? Please clarify randomization.

We have specified that individual randomization was used and provided additional details on the randomization procedure in the methodology.

8. How were 39.3% 3 years old and 60.7% 6 years old being recruited? Is this a proportional allocation?

The allocation was not proportional; the distribution of participants by age resulted from the natural recruitment process based on the availability of eligible children in each age group.

9. What about blinding? Who was blinded after assignment to interventions? How was it impossible to blind?

Blinding was not possible in this study because the children were aware of whether they were wearing the virtual reality glasses or not. As a result, both participants and healthcare personnel administering the vaccines were aware of the assigned intervention.

10. You have measured the outcome variable two times (first and second vaccination times). This can be considered a repeated measure. So, why not consider other advanced statistical analyses like GLM, GEE, etc.?

For greater clarity, we have provided medians of the variables, which have not influenced the previous results. While advanced statistical models were considered, we found that the current approach provided a clear and meaningful interpretation of the findings.

11. Please include how and when outcome variables were assessed.

We have provided additional details on the timing and methodology of the outcome assessments, specifying when pain, anxiety, and heart rate measurements were recorded.

12. The results part is good

Thank you.

13. Please try to improve the beginning part of the discussion. First, you have to mention your finding, interpret the finding, and then compare with others

We have restructured the Discussion section to begin with a clear statement of the main findings, followed by their interpretation and a comparison with previous literature, as recommended.

14. Some references needs URL and accessed date, e.g ref no. 3

We have reviewed and updated all references, ensuring that URLs and access dates are included where applicable.

---

## [Decision Letter · Decision Letter 1]

3 Mar 2025

PONE-D-24-20918R1Use of Virtual Reality in the reduction of pain after the administration of vaccines among children in Primary Care Centers in Central Catalonia: randomized clinical trial.PLOS ONE

Dear Dr. Josep Vidal-Alaball,

Thank you for submitting your manuscript to PLOS ONE. After careful consideration, we feel that it has merit but does not fully meet PLOS ONE’s publication criteria as it currently stands. Therefore, we invite you to submit a revised version of the manuscript that addresses the points raised during the review process.

 Be sure to:

Respond the reviewer feedback Follow the journal guidelines

Please submit your revised manuscript by  Apr 17 2025 11:59PM. If you will need more time than this to complete your revisions, please reply to this message or contact the journal office at plosone@plos.org . Please include the following items when submitting your revised manuscript:

We look forward to receiving your revised manuscript.

Kind regards,

Fadwa Alhalaiqa

Academic Editor

PLOS ONE

Journal Requirements:

Reviewers' comments:

Reviewer's Responses to Questions

**Comments to the Author**

1. If the authors have adequately addressed your comments raised in a previous round of review and you feel that this manuscript is now acceptable for publication, you may indicate that here to bypass the “Comments to the Author” section, enter your conflict of interest statement in the “Confidential to Editor” section, and submit your "Accept" recommendation.

Reviewer #1: (No Response)

2. Is the manuscript technically sound, and do the data support the conclusions?

Reviewer #1: (No Response)

3. Has the statistical analysis been performed appropriately and rigorously? 

Reviewer #1: (No Response)

4. Have the authors made all data underlying the findings in their manuscript fully available?

Reviewer #1: (No Response)

5. Is the manuscript presented in an intelligible fashion and written in standard English?

Reviewer #1: (No Response)

6. Review Comments to the Author

Reviewer #1: Tables 2 and 3: use decimal points instead of "comma". Be consistent with decimal numbers. The tables showed both 1 and 2 decimal numbers for percentages.

7. PLOS authors have the option to publish the peer review history of their article (what does this mean? ). If published, this will include your full peer review and any attached files.

**Do you want your identity to be public for this peer review?** For information about this choice, including consent withdrawal, please see our Privacy Policy .

Reviewer #1: No

---

## [Author Response · Author response to Decision Letter 1]

7 Mar 2025

Reviewer #1: Tables 2 and 3: use decimal points instead of "comma". Be consistent with decimal numbers. The tables showed both 1 and 2 decimal numbers for percentages.

Thank you for your feedback. We have made the requested changes to ensure consistency in decimal formatting. Specifically:

• Tables 1 and 2: We have replaced commas with decimal points for numerical values.

• Table 4: We have adjusted some results to maintain consistency in decimal formatting

Journal guidelines:

The abstract has been adjusted to 300 words taking into account the publication guidelines of the journal.

---

## [Decision Letter · Decision Letter 2]

31 Mar 2025

Use of Virtual Reality in the reduction of pain after the administration of vaccines among children in Primary Care Centers in Central Catalonia: randomized clinical trial.

PONE-D-24-20918R2

Dear Dr. Josep Vidal-Alaball,

We’re pleased to inform you that your manuscript has been judged scientifically suitable for publication and will be formally accepted for publication once it meets all outstanding technical requirements.

Kind regards,

Fadwa Alhalaiqa

Academic Editor

PLOS ONE

Additional Editor Comments (optional):

Reviewers' comments:

Reviewer's Responses to Questions

**Comments to the Author**

1. If the authors have adequately addressed your comments raised in a previous round of review and you feel that this manuscript is now acceptable for publication, you may indicate that here to bypass the “Comments to the Author” section, enter your conflict of interest statement in the “Confidential to Editor” section, and submit your "Accept" recommendation.

Reviewer #1: All comments have been addressed

2. Is the manuscript technically sound, and do the data support the conclusions?

Reviewer #1: (No Response)

3. Has the statistical analysis been performed appropriately and rigorously? 

Reviewer #1: (No Response)

4. Have the authors made all data underlying the findings in their manuscript fully available?

Reviewer #1: (No Response)

5. Is the manuscript presented in an intelligible fashion and written in standard English?

Reviewer #1: (No Response)

6. Review Comments to the Author

Reviewer #1: All my concerns are addressed.

7. PLOS authors have the option to publish the peer review history of their article (what does this mean? ). If published, this will include your full peer review and any attached files.

**Do you want your identity to be public for this peer review?** For information about this choice, including consent withdrawal, please see our Privacy Policy .

Reviewer #1: No

---

## [Editor Report · Acceptance letter]

PONE-D-24-20918R2

PLOS ONE

Dear Dr. Vidal-Alaball,

I'm pleased to inform you that your manuscript has been deemed suitable for publication in PLOS ONE. Congratulations! Your manuscript is now being handed over to our production team.

Kind regards,

on behalf of

Pro Fadwa Alhalaiqa

Academic Editor

PLOS ONE